# Enhancing Canarian Cockerel Meat with n-3 LC-PUFAs Through Echium and Linseed Oils: Implications on Performance and Meat Quality Attributes

**DOI:** 10.3390/foods14101730

**Published:** 2025-05-13

**Authors:** Jesús Villora, Alexandr Torres, Sergio Álvarez, Nieves Guadalupe Acosta, José Antonio Pérez, Covadonga Rodríguez

**Affiliations:** 1Departamento de Biología Animal, Edafología y Geología, Universidad de La Laguna, Avenida Astrofísico Francisco Sánchez s/n, 38206 La Laguna, Tenerife, Spain; ngacostaglez@ull.edu.es (N.G.A.); janperez@ull.edu.es (J.A.P.); 2Unidad de Producción Animal, Pastos y Forrajes en Zonas Áridas y Subtropicales, Instituto Canario de Investigaciones Agrarias, 38200 San Cristóbal de La Laguna, Tenerife, Spain; atorresk@icia.es (A.T.); salvarez@icia.es (S.Á.); covarodr@ull.edu.es (C.R.)

**Keywords:** local poultry, slow-growing chicken, *Echium* oil, meat quality, omega-3 PUFA, stearidonic acid

## Abstract

Interest in indigenous and dual-purpose chicken breeds for sustainable poultry farming is growing. Additionally, incorporating local feed resources into their diets may enhance the nutritional value of their products while reducing environmental impact. This study investigated the ability of *Echium* oil (EO), rich in stearidonic acid (SDA, 18:4n-3) compared to linseed oil (LO) and high in alpha-linolenic acid (ALA, 18:3n-3), to increase long-chain n-3 polyunsaturated fatty acids (LC-PUFAs) in breast meat. Sixty Canarian cockerels were fed for six weeks with diets supplemented with 1.5% soybean oil (SO), 1.5% LO, or 2% EO. Final body weight and carcass traits showed no significant differences among groups (*p* > 0.05). However, EO-fed birds exhibited slightly higher breast meat lightness (L*) than LO-fed ones (*p* < 0.05). Total lipid content and lipid class composition remained unchanged (*p* > 0.05). Both LO and EO increased eicosapentaenoic acid (EPA, 20:5n-3) compared to SO, with EO further enhancing SDA, 20:3n-3, 20:4n-3, docosapentaenoic acid (DPA, 22:5n-3), and docosahexaenoic acid (DHA, 22:6n-3), resulting in meat with a healthier thrombogenic index (TI). Importantly, EO inclusion up to 2% did not negatively impact meat sensory qualities. These findings suggest that EO outperforms LO in enriching poultry meat with beneficial n-3 LC-PUFAs and holds great potential for poultry production.

## 1. Introduction

N-3 long-chain polyunsaturated fatty acids (LC-PUFAs) such as eicosapentaenoic acid (EPA, 20:5n-3), docosapentaenoic acid (DPA, 22:5n-3), and docosahexaenoic acid (DHA, 22:6n-3) are crucial for proper animal growth and development [1]. Many vertebrates possess the enzymatic repertoire to biosynthesize n-3 LC-PUFAs from their dietary C18 precursor alfa-linolenic acid (ALA, 18:3n-3) through elongation and desaturation processes [2]. However, humans are relatively inefficient at synthesizing n-3 LC-PUFAs, which, therefore, must necessarily be obtained through the diet [3]. Marine fish are the primary source of n-3 LC-PUFAs for humans. Nevertheless, the sustainability challenges of aquaculture and the global depletion of most fish stocks due to overfishing make it essential to explore alternative sources of n-3 fatty acids (FAs) from terrestrial animals [4].

In recent decades, poultry meat consumption has risen globally due to its affordability as a source of protein and bioactive substances, including vitamins and antioxidants [5]. Poultry meat is considered the most cost-effective and sustainable terrestrial animal protein source because of its high efficiency in converting feed into meat [6]. Moreover, poultry has been regarded as a viable source of LC-PUFAs due to its inherent enzymatic capability for synthesizing these physiologically important FAs [7,8]. However, as current livestock diets are especially rich in linoleic acid (LA, 18:2n-6), poultry products such as meat and eggs are common sources of n-6 PUFAs, including LA and arachidonic acid (ARA, 20:4n-6) [9]. The biosynthesis of ARA and EPA from LA and ALA, respectively, involves the same desaturase and elongase enzymes, resulting in a competition between the two groups of PUFAs. Therefore, reducing the dietary n-6/n-3 ratio (LA/ALA) could increase the content of beneficial n-3 LC-PUFAs in poultry products [10].

The search for sources of n-3 LC-PUFAs has become a significant focus for the food industry. In this context, several strategies have been conducted to improve the nutritional quality of chicken meat, including feeding animals with vegetable oils richer in ALA such as LO [11,12,13]. However, the initial step in the biosynthesis of EPA and DHA from ALA to generate stearidonic acid (SDA, 18:4n-3) requires the Δ6 desaturase enzyme that is considered rate limiting [7]. Therefore, utilizing vegetable sources containing SDA may also represent a more effective strategy to enhance the levels of EPA, DPA, and DHA in the final product [14]. Although SDA is present in relatively low amounts in most plant oils, it is present at an important ratio in the Primulaceae, Cannabaceae, and Boraginaceae families, this last including the *Echium* genus [15]. In fact, *Echium* oil (EO) contains significant amounts of ALA, SDA, and the nutraceutical γ-linolenic acid (GLA, 18:3n-6), and moderate levels of LA [16,17].

The Canary Islands are recognized as a biodiversity hotspot, boasting over 680 endemic plant species and being the largest center for the *Echium* genus, with 28 endemic taxa [18]. Approximately 28% of the 13,000 plant and animal species in the Canaries are unique to the region, making it one of the world’s foremost areas of endemism [19]. As a result of the unique climate and location isolation, livestock animals such as chicken that were introduced during the XV and XVI centuries have generated, over time, a distinct breed with specific features [20]. Canarian native chickens are well adapted to their local environmental conditions and traditional breeding practices, linked to sustainable management models. In contrast to most genetically selected poultry breeds, the Canarian genotype is distinguished by its dual-purpose nature, well suited for fattening and egg laying [21]. Unlike commercial crossbreeds, local chicken breeds represent an important genetic heritage that must be conserved to maintain biodiversity and could be exploited in genetic selection programs to improve resilience to environmental stresses [22].

Previous studies [23,24,25] have examined the effects of SDA-rich diets on the lipid composition of chicken tissues. These investigations were conducted using fast-growing (FG) genotypes in intensive farming systems, where high growth rates prioritize muscle development and influence polyunsaturated fatty acids (PUFAs) accumulation [26]. In contrast, free-range systems favor animal welfare and sustainability, providing chickens with a better life quality [27]. While this approach may be seen as economically disadvantageous, it can represent added value to consumers by offering a more acceptable and sustainable method for poultry production [28]. Recent data published by Villora et al. [14] demonstrated that slow-growing Canarian chickens fed diets supplemented with EO exhibited increased expression of hepatic elongases, which correlated with higher n-3 LC-PUFAs levels in thigh meat. This research aims to complement these findings by investigating the growth, performance, FA composition, and sensory qualities of Canarian cockerel breast meat. Specifically, the study addresses two key issues: (1) whether a diet containing higher proportions of SDA and ALA is more effective than a richer one in ALA, and (2) whether the benefits are more pronounced in dual-purpose native breeds like Canarian chicken than in intensively reared broilers, based on available data.

## 2. Materials and Methods

### 2.1. Animals and Diets

The trial was carried out at the experimental farm of the Instituto Canario de Investigaciones Agrarias in Tenerife (Spain). Hatching eggs from the Canarian genotype were sourced from the Asociación La Campera Canaria (Tenerife, Spain). The eggs were incubated for 21 days (Masalles M240-I, Barcelona, Spain) and after hatching, the chicks were maintained all together in interior pens equipped with warm lamps and fed with commercial starting feed (Grupo Capisa, Tenerife, Spain). Birds were vaccinated against Marek and Newcastle diseases. After 4 weeks of life, animals were transferred to outdoor pens and fed with a commercial growing feed, mainly containing wheat meal, soy meal, and corn meal (Grupo Capisa). In week 8, sexing was performed and a total of 60 males were relocated according to weight in three pens (*n* = 20 birds per pen), with access to 25 m^2^ outdoor facilities. Animals were raised under free-range conditions with unrestricted provision of water and food. After 18 weeks of life, feed was changed to experimental diets. Birds from each pen received a different diet prepared by spraying 3 distinct vegetable oil mixtures to a common fat-reduced cereal base (Grupo Capisa), as detailed in Table 1. Three experimental groups were formed: Soybean oil diet (SO-d), supplemented with 1.5% SO and 1.5% beef tallow; Linseed oil diet (LO-d), supplemented with 1.5% LO and 1.5% beef tallow; *Echium* oil diet (EO-d), containing 2% *E. plantagineum* oil and 1% beef tallow. The diets were formulated to achieve the same n-6/n-3 ratio in the LO-d and EO-d (2.29 and 2.22, respectively), but a higher ratio of 10.83 in the SO-d. All the diets were isoenergetic, isoproteic, and isolipidic and were supplied ad libitum for 6 weeks. Across the experimental period, feed samples were collected (*n* = 3) to determine their FA composition (Table 1) and birds’ body weights were recorded weekly.

### 2.2. Sample Collection

Approximately 12 h prior to slaughter, feed was withdrawn, and animals were weighed. A total of 14 cockerels per group were slaughtered after 6 weeks of feeding trail at the slaughterhouse of the SADA p.a. Canarias SA Group (Tenerife, Spain), according to European regulations [29]. The carcasses were refrigerated for 24 h at 4 °C to determine carcass weight and dressing percentage. The breast, legs (drumstick + thigh), and wings were removed, weighed, and calculated as a percentage of the total carcass weight. The left breast samples were used to measure pH, color parameters, cooking loss, and shear force, while the right breast samples were vacuum-packed and kept at −20 °C until chemical composition and sensory analyses were conducted within two months. A portion of the right breasts were also transported to the laboratories of the NUTRAHLIPIDS group at the Departamento de Biología Animal, Edafología y Geología (Universidad de La Laguna, Tenerife, Spain) to determine their fat content, lipid classes, and FA composition.

### 2.3. Physicochemical Analysis

Breast meat pH was determined in triplicate using a penetration pH electrode by means of a pH meter GLP 21 (Crison Instruments SA, Barcelona, Spain). Breast meat color was measured immediately after skin removal. Color coordinates CIE, L* (lightness), a* (redness), and b* (yellowness) were measured on the breast surface using a CR-400 Chroma Meter (Minolta Ltd., Milton Keynes, UK) calibrated with a standard white tile. For each sample, three measurements were taken at the same anatomical position. Cooking losses were assessed according to the procedures described by Díaz et al. [30]. Shear force was measured using a TA-HD-Plus texture analyzer equipped with a Kramer Shear Cell (Stable Microsystems, Surrey, UK). Shear tests of cooked meat were conducted in triplicate by cutting cores (1 cm^2^ in cross-section and 3 cm long) parallel to the muscle fibers. Chemical analyses of the breast, including moisture, protein, and ash content, were carried out by Trouw Nutrition Masterlab (Madrid, Spain), an accredited facility for food product testing. All analytical measurements were performed using *n* = 14.

### 2.4. Lipid Composition

The total lipid (TL) or fat content of the diets (*n* = 3) and breast meat (*n* = 4) was extracted in chloroform/methanol (2:1, *v*/*v*) according to Folch et al. [31], with small modifications as described by Reis et al. [32].

Lipid classes (*n* = 4) were analyzed by high-performance thin layer chromatography (HPTLC) in a one-dimensional double-development system to separate polar and neutral lipids [33]. Lipid classes were identified by comparison to external lipid standards placed on the same HPTLC plate and quantified by calibrated densitometry using a dual wavelength flying spot scanner CAMAG TLC Visualizer (Camag, Muttenz, Switzerland), as described by Reis et al. [32].

Fatty acid methyl esters (FAMEs) (*n* = 4) were obtained from 1 mg of total lipids by acid-catalyzed transmethylation using toluene and 1% sulfuric acid in methanol (*v*/*v*) for 16 h at 50 °C [34]. For absolute quantification of FAs (mg FA per 100 mg), 5% (50 µg) of 19:0 was added, as an internal standard, to the lipid extracts prior to transmethylation. FAMEs were then purified by thin-layer chromatography (TLC) and subsequently separated and quantified using a TRACE-GC Ultra Gas Chromatograph (Thermo Scientific, Milan, Italy), as detailed by Galindo et al. [35]. FAMEs were identified by comparison with retention times of a commercial standard mixture (FAME Mix C4-C24 and PUFA N° 3 from menhaden oil, Supelco Inc., Bellefonte, PA, USA) and to a well characterized cod roe FAME. Further confirmation of FA identity was carried out by GC–MS (DSQ II; Thermo Electron Corp, Waltham, MA, USA) when necessary. Results are expressed as a percentage of total FAs for the diets, and as mg FA per 100 mg of tissue for the breast meat.

### 2.5. Nutritional Indices

Nutritional quality of breast was assessed for all dietary treatments in terms of muscle FA composition, calculating the atherogenic (AI) and thrombogenic indices (TI) [36] as well as the ratio between hypocholesterolemic and Hypercholesterolemic FA (hH) [37], as follows:AI = [12:0 + (4 × 14:0) + 16:0]/(∑MUFAs + ∑n-6 PUFAs + ∑n-3 PUFAs)TI = (14:0 + 16:0 + 18:0)/(0.5 × ∑MUFAs + 0.5 × ∑n-6 PUFAs + 3 × ∑n-3 PUFAs + n-3/n-6 ratio)hH = (18:1n-9 + 18:2n-6 + 20:4n-6 + 18:3n-3 + 20:5n-3 + 22:5n-3 + 22:6n-3)/(14:0 + 16:0)

### 2.6. Sensory Analysis

Sensory profiles of breast meat were analyzed by 9 trained panelists from the Instituto Canario de Investigaciones Agrarias with experience in the evaluation of meat profiles, following the instructions given by the norm ISO 8589 [38].

Samples of breast meat (*pectoralis major*) were unfrozen at 4 °C for 24 h, trimmed of any external connective tissue, cut into 2 cm × 2 cm pieces, and then wrapped into aluminum foil, coded with a three-digit random number. The pieces were cooked on a double-sided grill previously heated at 180 °C until reaching an internal temperature of 72 °C. Sample presentation was balanced to account for order and carryover effects [39]. One meat sample per dietary treatment (SO-d, LO-d and EO-d) was evaluated by each panelist. Mineral water and unsalted bread were provided for mouth rinsing between samples.

The score set included nine descriptors for meat associated with the intensity of odor, flavor, and texture in a 9-point intensity scale of 1 (very low) to 9 (very high) [40], while acceptance was scored using a 9-point hedonic scale.

Before the sensory evaluation, all panelists signed a written consent form to participate.

### 2.7. Statistical Analysis

Prior to analysis, normality and homoscedasticity within groups were confirmed. When necessary, variance-stabilizing transformations (arcsine and logarithm) were applied. When both assumptions were met, significant differences between treatments were determined using one-way ANOVA followed by Tukey’s HSD post hoc test. For non-homoscedastic data, Welch’s test followed by Dunnett’s T3 test was used. In the case of a non-normal distribution, the Kruskal–Wallis non-parametric test was applied, followed by pairwise comparisons using the Mann–Whitney test with Bonferroni correction. Principal Component Analysis (PCA) of PUFAs was carried out and factor scores were subsequently analyzed via ANOVA.

Results are presented as means and standard error of the mean (SEM). Statistical differences were considered significant at *p* < 0.05. All statistical analyses were carried out using the IBM^®^ SPSS Statistics 25.0 software package (IBM Corp., New York, NY, USA).

### 2.8. Ethics Statement

All animal procedures were conducted in accordance with the European Directive 2010/63/EU and the Spanish RD53/2013 on the protection of animals used for scientific purposes. In this study, the animals were reared and slaughtered at an official slaughterhouse following standard commercial farm practices without applying any experimental or invasive procedures beyond routine management.

## 3. Results

### 3.1. Body Weight and Carcass Characteristics

At the end of the experimental period, there were no significant differences in cockerel body weight (2.89–2.99 kg), carcass yield (2.02–2.18 kg), or dressing percentage (70.00–73.18%) among the dietary groups (Table 2). Likewise, the feed intake and the weights and yields of the main joints remained unchanged, regardless of the oil supplemented in the diet. Consequently, the FCR values remained stable between experimental groups, ranging from 5.09 to 5.36 over the full 6-week dietary trial.

### 3.2. Physical Characteristics and Chemical Composition

As shown in Table 3, breast meat pH (5.77–5.82) was not affected by the dietary treatment. By contrast, the breast meat of cockerels fed EO exhibited a slightly higher lightness (L*) compared to those fed LO. Dietary treatments did not significantly vary meat redness (a*) and yellowness (b*). Cooking loss (15.67–16.89%), meat tenderness measured through the shear force (24.46–27.48 N/cm^2^), and chemical composition were not significantly affected by the dietary inclusion of SO, LO, or EO (Table 3).

### 3.3. Lipid Classes Composition

The lipid class composition of Canarian cockerel breasts showed a balanced proportion of total polar and total neutral lipid fractions (Table 4). The lecithin phosphatidylcholine (PC, 20.20–23.88%), together with phosphatidylethanolamine (PE, 14.45–16.39%), were the most abundant phospholipids, while the major neutral lipid components were cholesterol (CHO, 16.14–17.70%) and triacylglycerols (TAGs, 10.43–18.42%).

### 3.4. Fatty Acid Composition

As displayed in Table 5, the saturated fatty acid (SFA) proportion of breast meat was not affected by the dietary composition (~30% of total FA), 16:0 being the most abundant FA within this group. Oleic acid (OLE, 18:1n-9) was consistently the highest monounsaturated fatty acid (MUFA, 16.52–20.77%) in all treatments. Although meat LA levels remained unchanged, EO, and more effectively LO, reduced ARA with respect to SO (22% and 33% reduction, respectively). In addition, LO and EO also diminished the meat content of C22 FAs derived from the elongation and desaturation of LA. By contrast, birds fed diets supplemented with LO and EO presented higher quantities of total n-3 PUFAs compared to those fed the SO-d (31.16 and 36.46 vs. 16.13 mg/100 g, respectively) mainly due to a greater concentration of ALA and EPA. Interestingly, EO additionally boosted meat proportions of C22 n-3 LC-PUFAs, DPA, and DHA. Other beneficial minor FAs such as GLA, 20:3n-6, SDA, and 20:4n-3 also showed higher concentrations in the EO-birds.

The PCA for PUFAs revealed that PC1 and PC2 accounted for 51.77 and 23.26% of the variance, respectively (Figure 1). PC1, which is highly positively related to n-3 PUFAs, including SDA (0.85), 20:4n-3 (0.90), EPA (0.80), DPA (0.90), and DHA (0.79), arranged the data into three distinct clusters.

### 3.5. Nutritional Indices

Key nutritional indicators such as AI (0.35–0.39) and hH proportion (2.40–2.68) remained unchanged regardless of the diet, while the dietary inclusion of EO led to a significant reduction in the TI (0.75 and 0.68 vs. 0.55; Table 6).

### 3.6. Sensory Analysis

The trained sensory panel did not find significant differences among treatments in either the intensity or the acceptance-assessed attributes of Canarian cockerel breast meat (Table 7).

## 4. Discussion

The dietary supplementation with SO, LO, or EO resulted in similar chicken growth performance, as the final live bird weights were all around 3 kg (Table 2). Although overall data on the Canarian chicken are limited, Torres et al. [41] evaluated the morphological characteristics of the local rooster and hen populations in the Canary Islands, establishing a mean weight of 3.5 kg in adult males. In this study, the carcass weight was approximately 2.1 kg, yielding a dressing percentage of 70–73%. This is notably higher than the 64.34% reported by Torres et al. [21] for male Canarian cockerels slaughtered at 15 weeks, likely because our measurements were taken at 24 weeks. LO and EO did not vary performance outcomes such as carcass weights and yields of main joints when compared to the SO-d. Our findings are consistent with those by Kitessa and Young [23] when using EO-supplemented diets in Australian broilers and also align well with other studies that reported no impact of dietary LO on carcass yields [42,43,44]. Similarly, no differences in animal weight, carcass yield, or yield of different meat cuts were evident when incorporating oils high in SDAs, such as SDA-enriched soybean oil [24] or ahiflower [25].

Breast meat pH at 24 h post-slaughter (pH_24_) remained fairly constant regardless of the dietary group (5.77–5.82; Table 3). These values are within the expected limits of 5.75 and 5.96 at the end of post-mortem process outlined by Castellini et al. [45]. Additionally, and in agreement with our results, previous research reported no difference in pH_24_ after the inclusion of LO in the chicken diet [46,47,48]. Likewise, Torres at al. [21] stated a mean pH_24_ value of 5.85 from Canarian cockerel meat fed with a commercial diet. Muscle pH is linked to various important attributes of meat quality, including color, cooking losses, and tenderness [49]. As reviewed by Wideman et al. [50], meat color is highly correlated with the amount of myoglobin in the muscle but is also influenced by pH and muscle type. There are contradictory results when studying the correlation between the dietary inclusion of n-3 rich oils and meat physical properties. While some studies have found no significant relationship between both factors [46,51,52], others have shown some association. For instance, Qi et al. [53] observed that reducing the n-6/n-3 ratio decreased breast muscle luminosity (L*). In our present research, there was no significant correlation between the dietary n-6/n-3 ratio and meat color parameters. However, the dietary inclusion of EO increased luminosity (L*) with respect to LO-d, resulting in lighter breast meat. This suggests that while the n-6/n-3 ratio itself may not directly influence meat color, the specific type of oil incorporated into the diet can differentially affect color characteristics.

The cooking loss values registered in our study are slightly lower than those reported by Torres et al. [21] for the Canarian genotype, but they are in line with those obtained in the same research for Les Blues, another dual-purpose breed, at 15 weeks of age. Consistently, and regardless of the dietary group, meat shear forces were barely lower than those by Torres et al. [21], despite their research using younger birds. This evidence contradicts the general trend described for different bird species where shear force typically increases with age since muscle protein content increases more sharply than other components, and the composition of the connective tissue becomes denser as the livestock mature [26,54,55]. However, and in agreement with our results, some studies did not find differences in drip losses [56] or shear force [48,52,57] after the inclusion of dietary sources rich in PUFAs.

Under our experimental conditions, the dietary FA profile did not vary the chemical composition of chicken breast meat, including moisture, protein, fat, and ash (Table 3). Our data confirm earlier findings that the dietary inclusion of LO did not significantly alter the protein content of birds’ meat [43], nor did the lipid content change with the incorporation of n-3 PUFA-rich sources such as SDA-enriched soy oil [58], linseed oil [44], or chia seeds [56]. Likewise, Kitessa and Young [23] indicated that intramuscular fat remained constant when broilers were fed 3 g/kg of EO. The slightly higher meat protein proportions (26.42–27.33%), and the less than half total fat content (0.57–0.61%) recorded in our experiment compared to those of Canarian cockerels raised under free-range conditions [21], might be attributed to a different age of slaughter [59]. In this context, the lipid class composition of the muscle was stable, independently of the dietary treatment (Table 4) with PC and PE as the dominant phospholipids [8,60], as both are integral essential components of biological membranes. To our knowledge, scientific data on the impact of omega-3-rich vegetable oils on the lipid class composition of poultry meat remain very limited.

The LA levels of breast meat were similar in the three groups of Canarian cockerels (44.46–52.49 mg/100 fresh meat; Table 5), despite the different dietary contents (~38% in the SO-d and ~29% in the LO and EO diets, Table 1). Nevertheless, dietary LO and EO reduced ARA and its elongation and desaturation products with respect to SO. In agreement with our results, Rymer et al. [58] and Elkin et al. [24] demonstrated that lowering dietary LA led to a reduction of ARA in chicken breast meat. A reduction in the ingestion of ARA is associated with reduced inflammatory responses and improved cardiovascular health [61]. In contrast, EO increased GLA and 20:3n-6 compared to the other experimental diets (Table 5). Sergeant et al. [62] highlighted the potential of GLA, especially when combined with n-3 LC-PUFAs, to mitigate inflammation and to improve symptoms of inflammatory diseases. Additionally, 20:3n-6 has been reported to produce bioactive compounds with anti-inflammatory, vasodilatory, and anti-neoplastic effects while also lowering blood pressure and inhibiting smooth muscle cell proliferation [63]. Hence, the FA profile of breast meat from birds fed EO might be considered the most beneficial one for human health.

Previous studies found no evidence that birds converted SDA to DHA more efficiently than ALA [23,24,25,58]. A regulatory step after the initial hepatic Δ6 desaturation, possibly the second Δ6 step in the Sprecher pathway (desaturation of 24:5n-3 to 24:6n-3), might be limiting the production of DHA. In this study, chickens fed the EO-d exhibited superior efficiency versus LO in promoting the accumulation of n-3 LC-PUFAs, including DPA and DHA (7.03 vs. 9.41 mg/100 g, respectively, Table 5), indicating a more effective role in enriching poultry meat. The discrepancies between previous studies and the present research may be due to genetic selection. While earlier studies focused on FG broilers, our study examined the Canarian breed, a slow-growing (SG) and dual-purpose local genotype. In this regard, Boschetti et al. [64] confirmed that the SG genotype exhibited enhanced expression of FADS1 and FADS2 genes and higher Δ6 and Δ5 desaturase activity. Additionally, recent studies by Cartoni-Mancinelli et al. [65] have demonstrated higher expression of FADS2 and increased Δ6 desaturase activity in SG (Leghorn) compared to FG (Ross 308), enabling more efficient synthesis of n-3 LC-PUFAs. These findings highlight the role of genetic factors in LC-PUFA synthesis and suggest that selecting slow-growing genotypes with higher elongase and desaturase activity could be a promising strategy for improving the nutritional quality of poultry products.

Most health organizations recommend adults to daily consume between 250 and 1000 mg of EPA + DHA for overall health and well-being, although specific guidelines for different ages and physiological groups still require further research [66]. In Europe, current legislation requires that foods must contain a minimum of 40 mg per 100 g and 100 kcal of EPA + DHA to be considered a source of omega-3 [67]. Breast meat from chicken fed the EO-supplemented diet contained approximately 12.5 mg of EPA and DHA per 100 g of fresh tissue, which falls below the threshold required to be regarded as an omega-3 source (Table 5). Nevertheless, chicken breast meat might still represent an important dietary contribution of n-3 LC-PUFAs, particularly for populations with limited access to marine products. Moreover, it is important to note that other n-3 LC-PUFAs such as DPA were the predominant n-3 FAs in the breast meat of EO-supplemented birds (14.4 mg/100 g). DPA possesses some beneficial and potentially unique properties such as reducing the expression of inflammatory genes, preventing angiogenesis, and inhibiting platelet aggregation more effectively than EPA and DHA [68]. In concordance, the inclusion of EO resulted in breast meat with a lower TI compared to SO-d and LO-d (Table 6), which has a protective action against coronary heart diseases [69].

The inclusion of vegetable oils richer in n-3 PUFAs did not lead to variations in the sensory perception evaluated by the trained panel compared to samples from animals fed with higher n-6 FA proportions (Table 7). Odor and flavor, alongside texture, are key sensory quality attributes for consumers when determining the acceptability of chicken meat [70]. Although LO and EO diets resulted in slightly lower scores for breast odor and flavor compared to the SO-d, these differences were not statistically significant. Similarly, Rymer et al. [60] reported that slight differences perceived in the breast meats of birds fed SDA-enriched SO were associated with the texture and appearance of the meat, but not its flavor, aroma, or aftertaste. Consistent with our findings, other authors observed no significant variations in the sensory quality attributes of chicken meat due to the dietary inclusion of different LO levels [11,40,71,72]. It is noteworthy that all groups received high scores across the analyzed variables. Of particular interest is the appearance, which consistently scored close to 7 points for all diets. These values are moderately higher than those reported by Torres et al. [21], not only for the Canarian chicken genotype but also for other breeds evaluated in that study, such as Les Blues and Dominant Red Barred. Notably, undesirable odors or unusual aromas were not detected by the inclusion of dietary EO. These results suggest that, from a sensory perspective, the use of experimental diets, including those incorporating EO, could be suitable for poultry production.

## 5. Conclusions

This study demonstrates that EO offers significant benefits over LO in modulating the lipid metabolism in Canarian cockerels, a slow-growing, dual-purpose genotype. Both oils enhanced n-3 FA levels in breast meat compared to SO, but EO was notably more effective in increasing LC-PUFAs, such as EPA, DPA, and DHA, due to its higher SDA content, which bypasses the first limiting Δ6 desaturase enzyme step. Although breast n-3 LC-PUFA levels did not meet the regulatory threshold for labeling as an omega-3 source, the dietary incorporation of up to 2% EO resulted in a favorable FA composition of the meat without affecting performance, meat quality, and sensory attributes. The present approach supports agriculture using local breeds and feed sources, reducing the environmental footprint of poultry production while preserving biodiversity, making *Echium plantagineum* oil or that obtained other *Echium* species a promising option for nutritionally enhanced and eco-friendly poultry farming.

## Figures and Tables

**Figure 1 foods-14-01730-f001:**
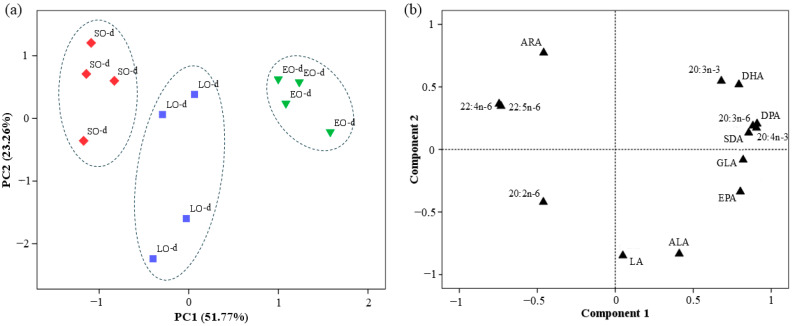
Principal component analysis (PCA) of PUFA of TL extract from breast meat of Canarian cockerel. (**a**) Factors score plot for PC1 and PC2. [(◆) SO-d = soy oil diet; (■) LO-d = linseed oil diet; (▼) EO-d = *Echium* oil diet]. Dashed line (---) represents different clusters for PC1 (*p* < 0.05). (**b**) Component loading plot for the PCA that illustrates the correlation between each individual PUFA and the principal components PC1 and PC2. LA, 18:2n-6; GLA, 18:3n-6; ARA, 20:4n-6; ALA, 18:3n-3; SDA, 18:4n-3; EPA, 20:5n-3; DPA, 22:5n-3; DHA, 22:6n-3.

**Table 1 foods-14-01730-t001:** Ingredients (%), nutrition facts, and main fatty acid composition (% of total fatty acids) of the 3 experimental diets.

Item	SO-d	LO-d	EO-d
Ingredients			
Soy oil	1.5	-	-
Linseed oil	-	1.5	-
*Echium plantagineum* oil	-	-	2
Beef tallow	1.5	1.5	1
Basal mixture ^1^	97	97	97
Calculated nutrition facts			
Energy (Kcal/100 g)	293.47	292.64	292.30
Crude carbohydrates (%)	41.13	41.13	41.13
Crude protein (%)	19.53	19.53	19.53
Moisture (%)	12.56	12.56	12.56
Crude ash (%)	4.66	4.66	4.66
Crude fiber (%)	3.16	3.16	3.16
Calcium (%)	0.69	0.69	0.69
Phosphorus (%)	0.52	0.52	0.52
Analyzed nutrition facts			
Crude fat (%)	4.77	4.68	4.64
Main fatty acid composition			
∑SFA	24.57	23.43	21.74
16:0	16.95	15.84	14.98
18:0	6.26	6.17	5.38
∑MUFA	32.83	32.27	29.36
18:1n-9	27.11	27.36	24.72
18:1n-7	2.88	2.49	2.21
∑n-6 PUFA	38.04	29.51	32.49
18:2n-6	37.77	29.15	28.59
18:3n-6	0.10	0.16	3.73
∑n-3 PUFA	3.51	12.89	14.60
18:3n-3	3.34	12.68	10.83
18:4n-3	0.12	0.16	3.71
∑n-6 LC-PUFA	0.17	0.20	0.17
∑n-3 LC-PUFA	0.05	0.05	0.06
n-6/n-3	10.83	2.29	2.22

SO-d, soy oil diet; LO-d, linseed oil diet; EO-d, *Echium* oil diet. SFA, saturated fatty acids; MUFA, monounsaturated fatty acids; PUFAs, polyunsaturated fatty acids; LC-PUFAs, long-chain (≥C20) polyunsaturated fatty acids. Totals include other minor components not shown. ^1^ Basal mixture composed of soy meal, wheat meal, corn meal, barley meal, phosphate, calcium carbonate, liquid methionine, lysine, salt sodium bicarbonate, axtraXB^®^, fysal^®^, and theonine (Capisa group).

**Table 2 foods-14-01730-t002:** Body weight and carcass characteristics of Canarian cockerels fed diets supplemented with soy, linseed, and *Echium* oils.

	SO-d	LO-d	EO-d	SEM	*p*-Value
Final body weight (kg)	2.99	2.90	2.89	0.057	0.767
Carcass weight (kg)	2.18	2.10	2.02	0.048	0.401
Dressing percentage	70.00	73.18	72.98	0.818	0.301
Feed intake (g/bird/day)	120.77	126.10	126.48	2.653	0.641
Commercial cuts ^1^					
Breast (g)	353.14	348.71	363.92	10.393	0.838
Breast (%)	12.28	12.04	12.13	0.313	0.463
Drumstick + thigh (g)	748.93	764.50	798.38	16.853	0.491
Drumstick + thigh (%)	26.03	26.73	26.68	0.721	0.936
Wings (g)	230.93	236.57	239.31	5.898	0.717
Wings (%)	8.02	8.28	8.03	0.162	0.399

SO-d, soy oil diet; LO-d, linseed oil diet; EO-d, *Echium* oil diet. ^1^ Commercial cuts percentages are expressed relative to carcass weight.

**Table 3 foods-14-01730-t003:** Physical characteristics and chemical composition of breast meat of Canarian cockerels fed diets supplemented with soy, linseed, and *Echium* oils ^1^.

	SO-d	LO-d	EO-d	SEM	*p*-Value
pH	5.82	5.78	5.77	0.033	0.472
Meat color					
L*	54.34 ^ab^	52.08 ^a^	55.82 ^b^	0.680	0.048
a*	2.18	2.47	2.03	0.137	0.414
b*	−1.35	−2.00	−1.51	0.273	0.560
Cooking loss (%)	15.67	16.02	16.89	0.551	0.665
Shear force (N/cm^2^)	24.46	26.80	27.48	1.038	0.481
Moisture	70.94	71.24	71.86	0.185	0.785
Protein	27.33	27.05	26.42	0.205	0.179
Fat	0.60	0.61	0.57	0.021	0.766
Ash	1.13	1.10	1.15	0.015	0.446

SO-d, soy oil diet; LO-d, linseed oil diet; EO-d, *Echium* oil diet. ^1^ Parameters determined 24 h post-mortem. L* (lightness), a* (redness), and b* (yellowness). ^a,b^ Means with different superscripts within the same row are significantly different (*p* < 0.05).

**Table 4 foods-14-01730-t004:** Total lipid content (% fresh weight) and lipid class composition (% of total lipid) of breast meat of Canarian cockerels fed diets supplemented with soy, linseed, and *Echium* oils.

	SO-d	LO-d	EO-d	SEM	*p*-Value
SM	2.25	1.74	1.58	0.128	0.072
PC	23.88	20.98	20.20	0.872	0.105
PS	3.09	2.45	2.76	0.135	0.158
PI	4.50	3.55	4.39	0.214	0.144
PG	4.19	3.86	3.19	0.183	0.059
PE	16.39	16.28	14.45	0.461	0.156
TPL	54.30	48.86	46.57	1.563	0.077
MAG	5.29	4.37	4.48	0.195	0.099
DAG	2.34	2.79	2.74	0.193	0.629
CHO	17.70	16.14	16.52	0.667	0.655
FFA	7.79	6.73	8.14	0.543	0.590
TAG	10.43	18.42	17.10	2.278	0.338
SE	2.14	2.68	4.46	0.469	0.098
TNL	45.70	51.14	53.43	1.563	0.077
Total lipid	0.60	0.61	0.57	0.021	0.766

Values are means (*n* = 4). SO-d, soy oil diet; LO-d, linseed oil diet; EO-d, *Echium* oil diet; SM, sphingomyelin; PC, phosphatidylcholine; PS, phosphatidylserine; PI, phosphatidylinositol; PG, phosphatidylglycerol; PE, phosphatidylethanolamine; TPLs, total polar lipids; MAGs, monoacylglycerols; DAGs, diacylglycerols; CHO, cholesterol; TAGs, triacylglycerols; SEs, sterol esters; TNLs, total neutral lipids.

**Table 5 foods-14-01730-t005:** Main fatty acid composition and total fatty acid content (mg/100 g of fresh weight) of breast meat of Canarian cockerels fed diets supplemented with soy, linseed, and *Echium* oils.

	SO-d	LO-d	EO-d	SEM	*p*-Value
∑SFA	114.53	127.80	112.58	7.728	0.723
16:0	71.59	83.19	70.29	5.482	0.610
18:0	39.87	40.65	38.72	2.073	0.941
∑MUFA	76.82	100.13	81.02	8.630	0.547
16:1 ^1^	2.72	5.45	3.87	0.813	0.426
18:1n-9 (OLE)	62.36	83.01	67.39	7.141	0.513
18:1n-7	10.28	9.99	7.92	0.693	0.350
∑n-6 PUFA	110.70	96.38	101.74	4.278	0.940
18:2n-6 (LA)	44.46	52.49	47.59	4.453	0.794
18:3n-6 (GLA)	0.10 ^a^	0.27 ^a^	2.05 ^b^	0.325	0.016
20:2n-6	1.05	0.91	0.86	0.056	0.370
20:3n-6	1.81 ^a^	1.98 ^a^	4.40 ^b^	0.374	0.023
20:4n-6	55.72 ^b^	37.28 ^a^	43.54 ^ab^	2.728	0.003
22:4n-6	5.52 ^b^	2.57 ^a^	2.44 ^a^	0.479	0.023
22:5n-6	2.02 ^b^	0.88 ^a^	0.86 ^a^	0.177	0.001
∑n-3 PUFA	16.13 ^a^	31.16 ^b^	36.46 ^b^	2.691	0.001
18:3n-3 (ALA)	0.92 ^a^	12.03 ^c^	7.29 ^b^	1.463	0.001
18:4n-3 (SDA)	nd	nd	1.70	0.267	0.001
20:3n-3	0.19 ^a^	0.12 ^a^	0.53 ^b^	0.074	0.030
20:4n-3	nd	nd	0.41	0.061	0.001
20:5n-3 (EPA)	0.50 ^a^	2.28 ^b^	2.77 ^b^	0.327	0.018
22:5n-3 (DPA)	7.64 ^a^	9.68 ^a^	14.35 ^b^	0.904	0.001
22:6n-3 (DHA)	6.87 ^a^	7.03 ^a^	9.41 ^b^	0.437	0.010
∑n-6 LC-PUFA	66.13 ^b^	43.62 ^a^	52.10 ^ab^	3.226	0.015
∑n-3 LC-PUFA	15.21 ^a^	19.13 ^a^	27.46 ^b^	1.555	0.001
n-6/n-3	6.92 ^b^	3.13 ^a^	2.79 ^a^	0.592	0.001
Total FA	377.26	399.54	377.85	19.460	0.889
EPA+DHA	7.38 ^a^	9.32 ^b^	12.18 ^c^	0.647	0.001
EPA+DPA+DHA	15.02 ^a^	19.01 ^b^	26.52 ^c^	1.525	0.001

Values are means (*n* = 4). SO-d, soy oil diet; LO-d, linseed oil diet; EO-d, *Echium* oil diet; SFAs, saturated fatty acids; MUFAs, monounsaturated fatty acids; PUFAs, polyunsaturated fatty acids; LC-PUFAs, long-chain (≥C20) polyunsaturated fatty acids; nd, not detected; FAs, fatty acids. Totals include other minor components not shown. ^1^ 16:1 Contains n-7 and n-9 isomers. ^a,b,c^ Means with different superscripts within the same row are significantly different (*p* < 0.05).

**Table 6 foods-14-01730-t006:** Atherogenic index (AI), thrombogenic index (TI), and hypocholesterolemic/Hypercholesterolemic fatty acids ratio (hH) of breast meat of Canarian cockerels fed diets supplemented with soy, linseed, and *Echium* oils.

	SO-d	LO-d	EO-d	SEM	*p*-Value
AI	0.37	0.39	0.35	0.010	0.124
TI	0.75 ^b^	0.68 ^b^	0.55 ^a^	0.029	0.002
hH	2.48	2.40	2.68	0.056	0.097

SO-d, soy oil diet; LO-d, linseed oil diet; EO-d, *Echium* oil diet. AI, atherogenic index; TI, thrombogenic index; hH, hypocholesterolemic to Hypercholesterolemic fatty acids ratio. ^a,b^ Means with different superscripts within the same row are significantly different (*p* < 0.05).

**Table 7 foods-14-01730-t007:** Sensory analysis of breast meat of Canarian cockerels fed diets supplemented with soy, linseed, and *Echium* oils.

	SO-d	LO-d	EO-d	SEM	*p*-Value
Intensity					
Appearance ^1^	7.00	6.78	7.00	0.074	0.442
Odor	5.44	5.00	4.56	0.302	0.529
Off-odor	0.33	0.67	0.78	0.308	0.797
Flavor	4.33	4.11	3.44	0.327	0.454
Off-flavor	0.00	0.67	0.33	0.214	0.340
Hardness	4.83	4.19	4.21	0.202	0.389
Elasticity	3.97	3.42	3.20	0.224	0.365
Juiciness	3.97	3.52	3.64	0.201	0.653
Oiliness	1.44	1.64	1.67	0.170	0.760
Acceptance ^2^					
Odor	6.78	6.33	6.11	0.234	0.503
Flavor	6.33	5.89	5.44	0.289	0.496
Texture	5.89	6.11	6.11	0.223	0.973
Overall	6.33	5.67	6.11	0.196	0.368

SO-d, soy oil diet; LO-d, linseed oil diet; EO-d, *Echium* oil diet. ^1^ Panellist (n = 9) evaluated the attributes with a 9-point intensity scale and then rated the ^2^ acceptance using a 9-point hedonic scale.

## Data Availability

The original contributions presented in this study are included in the article. Further inquiries can be directed to the corresponding author.

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
