# Peer review of "Enhancing Canarian Cockerel Meat with n-3 LC-PUFAs Through Echium and Linseed Oils: Implications on Performance and Meat Quality Attributes"

_foods, 2025, doi:10.3390/foods14101730_

Round 1

Reviewer 1 Report

Comments and Suggestions for Authors

The article entitled "Enhancing Canarian Cockerel Meat with n-3 LC-PUFA through Echium and Linseed Oils: Implications on performance and meat quality attributes" provides interesting information on the benefits of supplementing Echium oil on a specific breed of birds. The article is well written. Materials and methods, as well as results, are well depicted. I have a few comments and suggestions that may help to improve the manuscript. I am attaching a file indicating these comments and suggestions. 

Author Response

Summary                       
We sincerely thank you for your thoughtful and constructive comments. Your detailed feedback has been invaluable in helping us improve the quality and clarity of our manuscript.

General comment: The article entitled "Enhancing Canarian Cockerel Meat with n-3 LC-PUFA through Echium and Linseed Oils: Implications on performance and meat quality attributes" provides interesting information on the benefits of supplementing Echium oil on a specific breed of birds. The article is well written. Materials and methods, as well as results, are well depicted. I have a few comments and suggestions that may help to improve the manuscript. I am attaching a file indicating these comments and suggestions. 
Response: Thank you very much for your positive and encouraging feedback on our manuscript.

Comment 1: (Line 113). Authors should provide an explanation why for EO the proportion of supplemented oil was higher than control groups (SO and LO). It should be tested at the same proportion to properly compare its effect.
Response: Thank you very much for this observation. We selected the specific dietary inclusion levels of linseed oil (LO) and Echium oil (EO) (1.5% vs 2.0%, respectively) to ensure that both experimental diets had an identical n-6/n-3 ratio −that has been previously demonstrated to impact meat nutritional quality− without changing their total lipid content. This experimental design allowed us to compare the differential effects of diets with the same omega-6 to omega-3 balance, but differed notably in their individual fatty acid profiles due to the unique composition of EO, particularly high in the proportion of stearidonic acid (SDA). Additional clarifying information has been now included in the manuscript (lines 118-119).

Comment 2. (Table 1). Is the fatty acid composition of the hole diets? or only of the oils?
Response: The composition shown in Table 1 refers to the dietary formulations, as already stated in the table caption.

Comment 3. (Lines 126-127). I suggest to move this phrase to the beginning of this paragraph.
Response: Following your suggestion, the phrase has been moved to the beginning of the paragraph (now line 127-128).

Comment 4. (Line 131). Indicate for how long.
Response: The timing has been included in the sentence (line 135).

Comments 5, 6 and 7 (Lines 218-221). Some parts of the text have been removed.
Response: Following your suggestion, the crossed out text has been removed from the MS.

Comment 8. (Line 314). Only darker than EO supplemented, not with SO.
Response: The referee is right and the sentence has been modified according to your suggestion (lines 323-325)

Comment 9. (Line 314). No differences were found.
Response: The referee is right and the sentence has been modified accordingly (lines 323-325)

Comment 10. (Lines 350-351). Indicate the table.
Response: Done. (Now line 362)

Comment 11. (Lines 361-363). Indicate the table.
Response: Done. (Now line 374)

Comment 12. (Lines 379-381). Indicate the table.
Response: Done. (Now line 393)

Comment 13. (Lines 389-390). Indicate the table.
Response: Done. (Now line 400)

Reviewer 2 Report

Comments and Suggestions for Authors

Abstract.

  1. Please add the levels of oils used in the study
  2. Please add more findings in the abstract 
  3. In my opinion it is not recommended to use sustainable poultry production herein in Ln 26 as the cost maybe higher. Please revise otherwise please add the costs of each experimental group 

2. Introduction is well written 

3. Materials and methods lack the accuracy such as the diets. Please add the whole ingredients and please add the description of nutrition facts calculated or analyzed findings 

4. Results 

  1. I'm not sure why values of yellowness are in minus values, please discuss
  2. Table 2 please add Feed intake and FCR
  3. Please discuss the protein contents is higher in the breast meat 27% but in the basal diet was 19%. In my opinion it is difficult to get this result 

Please add units to all data

Author Response

Summary                       
We would like to express our sincere gratitude for your constructive feedback and valuable insights. Your careful review and thoughtful suggestions have been extremely helpful in refining our manuscript.

Comment 1: (Abstract). Please add the levels of oils used in the study
Response: Done (lines 20-21).

Comment 2. (Abstract). Please add more findings in the abstract
Response: We do agree with the reviewer on the importance of including as many relevant findings as possible in the Abstract to provide a clear overview of the study. Accordingly, we have now included a new text, respecting the journal's 200 word-limit (lines 26-27).

Comment 3. (Abstract). In my opinion it is not recommended to use sustainable poultry production herein in Ln 26 as the cost maybe higher. Please revise otherwise please add the costs of each experimental group
Response: Thank you very much for your comment. We agree that the term "sustainable" may lead to confusion if it is not properly clarified. In our manuscript, we used "sustainable poultry production" not in reference to economic sustainability, but rather in terms of using local forage resources and promoting the use of native livestock breeds — both key aspects of environmental and genetic resource sustainability. However, we understand your concern and, to avoid ambiguity, we have removed the term “sustainable” from the sentence.

Unfortunately, it is not possible to perform an economic evaluation of the diets, as this was not the objective of the present study, which focused specifically on the effects of Echium and linseed oil supplementation on the nutritional profiles, meat quality, and performance of Canarian cockerels.

Comments 4. Introduction is well written
Response: Thank you very much for your positive comment on the Introduction section.

Comment 5. Materials and methods lack the accuracy such as the diets. Please add the whole ingredients and please add the description of nutrition facts calculated or analyzed findings
Response: Thank you very much for your comment. We have now added the whole list of ingredients for the experimental diets in Table 1 caption. Additionally, we have clarified which nutritional parameters were calculated, and which were analysed, to provide greater accuracy and transparency in the Materials and Methods section.

Comment 6. (Results). I'm not sure why values of yellowness are in minus values, please discuss
Response: While the b* parameter in the CIELAB system typically indicates yellowness when positive and blueness when negative, it is important to note that negative b values are obtained in certain types of poultry meat and skin, especially in breeds or conditions where pigmentation is low.

For instance, Mueller et al. (2011) reported b* values of -0.28 in the thigh meat of the Rozz PM3 chicken breed, and even more pronounced negative b* values in the skin of different chicken breeds, such as -1.64 (Sasso 51), -1.40 (Lohmann Dual), -2.13 (Belgian Maliness), and -2.96 (Schweizerhuhn). These results indicate that, under certain conditions such as low carotenoid deposition or specific genetic traits, the meat or skin may show slightly bluish or very pale tones that are reflected in negative b* values.

Reference:
- Mueller, S.; Kreuzer, M.; Siegrist, M.; Mannale, K.; Messikommer, R.E.; Gangnat I.D.M. Carcass and Meat Quality of Dual-purpose Chickens (Lohmann Dual, Belgian Malines, Schweizerhuhn) in Comparison to Broiler and Layer Chicken Types. Poult. Sci. 2011, 97, 3325-3336.

Comment 7. (Results). Table 2 please add Feed intake and FCR
Response: Thank you very much for your valuable comment. We have now added the Feed Intake data to Table 2, as requested. In addition, the Feed Conversion Rate (FCR) has been calculated based on the increment of the total biomass across the experimental period for each dietary group (20 birds per group) and the total feed intake recorded for each group. We have added a new text about FCR in the Results section of the manuscript (lines 228-229).

Comment 8. (Results). Please discuss the protein contents is higher in the breast meat 27% but in the basal diet was 19%. In my opinion it is difficult to get this result.
Response: We appreciate your observation and the opportunity to clarify this point. This apparent discrepancy is explained by the biological efficiency of protein deposition in poultry. Similar findings have been widely reported in the literature, e.g., in the studies by Sirri et al., (2011) and Torres et al., (2019), showing that standard broiler breast meat typically reaches these protein levels (22.4-25.1%) regardless of dietary crude protein being in the 18–20% range.

References:
- Sirri, F.; Castellini, C.; Bianchi, M.; Petracci, M.; Meluzzi, A.; Franchini, A. Effect of Fast-, Medium- and Slow-Growing Strains on Meat Quality of Chickens Reared under the Organic Farming Method. Animal. 2011, 5, 312-9
- Torres, A.; Muth, P.C.; Capote, J.; Rodríguez, C.; Fresno, M.; Valle-Zárate, A. Suitability of Dual-Purpose Cockerels of 3 Different Genetic Origins for Fattening under Free-Range Conditions. Poult. Sci. 2019, 98, 6564–6571.

Round 2

Reviewer 2 Report

Comments and Suggestions for Authors

Thanks